# DNA Damage and Repair in Eye Diseases

**DOI:** 10.3390/ijms24043916

**Published:** 2023-02-15

**Authors:** Joanna Sohn, Sang-Eun Lee, Eun-Yong Shim

**Affiliations:** 1Department of Molecular Medicine, The University of Texas Health Science Center at San Antonio, 7703 Floyd Curl Drive, San Antonio, TX 78229, USA; 2Keystone School, 119 E. Craig Pl., San Antonio, TX 78212, USA

**Keywords:** DNA damage and repair, diabetic retinopathy, age-related macular degeneration, glaucoma

## Abstract

Vision is vital for daily activities, and yet the most common eye diseases—cataracts, DR, ARMD, and glaucoma—lead to blindness in aging eyes. Cataract surgery is one of the most frequently performed surgeries, and the outcome is typically excellent if there is no concomitant pathology present in the visual pathway. In contrast, patients with DR, ARMD and glaucoma often develop significant visual impairment. These often-multifactorial eye problems can have genetic and hereditary components, with recent data supporting the role of DNA damage and repair as significant pathogenic factors. In this article, we discuss the role of DNA damage and the repair deficit in the development of DR, ARMD and glaucoma.

## 1. Eye Diseases

Chronic ocular pathologies such as diabetic retinopathy (DR), age-related macular degeneration (ARMD) and glaucoma are leading causes of blindness worldwide [1]. Indeed, patients with these conditions are predominantly referred by ophthalmologists to low-vision rehabilitation services due to vision impairments [2,3]. Hence, further understanding of the underlying mechanism of pathology is crucial in the development of novel therapeutics for preventing irreversible vision loss in these patients. DNA damage, in particular, is thought to play an important role in the progression of these conditions. The purpose of this review is to highlight different mechanisms of DNA damage and repair pertinent to the progression of ocular pathologies and their roles in short- and long-term treatment outcomes.

## 2. DNA Damage and Repair Mechanisms

Environmental and metabolic agents such as reaction oxygen species (ROS), UV radiation, alkylating agents, and heterocyclic aromatic amines damage DNA, causing the loss of genetic information, and can lead to cell death and aging if not repaired accurately and in a timely manner (reviewed in [1,2]). DNA damage also induces constellations of mutations such as base changes, single-base insertions and deletions (indels), short-sequence insertions and deletions, and larger chromosomal rearrangements that are common to many human diseases [2,3,4]. Importantly, the location and the functions of the eye make it particularly susceptible to exogenous DNA damage such as radiation and chemical exposure, and several DNA repair defects are associated with impaired eye functions [5,6]. Naturally, cells evolve multiple mechanisms to detect and eliminate DNA lesions (so called DNA repair) to avoid harmful consequences to cells and organisms and to sustain genetic integrity (Figure 1). The defects in DNA damage repair and signaling thus predispose humans to a host of diseases and accelerated aging [1]. We will briefly outline below the types of DNA damage most relevant to eye diseases and the basic framework of repair mechanisms and pathways responsible for removing these lesions.

*Base excision repair (BER)*: ROS or alkylating agents produce DNA base damages that can trigger base mispairing and substitutions [7]. ROS is also linked to dysfunctional mitochondria and cellular stress to hyperglycemia, the common conditions associated with eye diseases [8].

BER is a specialized repair pathway that senses these base lesions and cleaves the glycosidic bond of a modified base by one of several glycosylases, producing the nucleotides without a base, also known as the apyrimidinic/apurinic (AP) site, as the key repair intermediates (reviewed in [9]). The AP site is subsequently cleaved by the AP endonuclease and replaced with a correct nucleotide by DNA polymerase β. Alternatively, the repair DNA synthesis could extend beyond a single nucleotide and replace longer stretches of DNA at the lesion [10,11]. The late steps of BER also share mechanistically with that of single-strand break repair and is tightly coupled to cellular metabolic and energy status [12]. Since oxidative stress is the primary DNA lesion type in eye diseases, both hyperactive or reduced BER is implicated in DR [13,14]. Polymorphism of BER genes, MUTYH and hOGG1 are also associated with age-associated macular degeneration [15].

*Nucleotide excision repair (NER)*: The vision depends on light that is transmitted through the cornea and reaches retina [16]. UV radiation and UV mimetic agents, however, induce pyrimidine dimers and bulky DNA adducts, distorting the DNA structure and impeding faithful DNA replication and transcription (reviewed in [17]). NER is responsible for sensing such DNA distorting lesions, excises both sides of the aberrantly modified bases by two DNA endonucleases (ERCC1/XPF and XPG) and catalyzes resynthesis of the excised DNA sequence [18]. NER has two different variants that differ primarily at the lesion sensing step. Transcription coupled NER (TC-NER) links DNA lesion sensing and subsequent repair events to the transcription process and thereby focuses on eliminating DNA damage on DNA encoding transcripts [19]. Alternatively, global genomic-NER (GG-NER) recognizes DNA lesions anywhere in the genome using the DDB/XPC/hHR23B complex [20]. NER defects cause xeroderma pigmentosum, the recessive human genetic disorder with extreme UV sensitivity, cancer predisposition and ocular manifestation [21].

*Double strand break repair (DSBR)*: Ionizing radiation and radiomimetic agents induce DNA double-strand break (DSB), the most severe types of DNA damage and a major threat to the survival of cells and host organisms (reviewed in [22]). Unrepaired DSBs are also capable of inducing chromosome breakage and translocation, and the production of fusion genes, the hallmarks of cancer cells. Even though the formation of DSBs is not inherent to eyes, such damage could arise as the secondary lesions if the original ones persist and are further modified by cellular events such as DNA replication [23]. In all eukaryotic cells, the efficient elimination of DSBs relies on two evolutionary conserved mechanisms: homologous recombination (HR) and non-homologous end joining (NHEJ). HR begins with the formation of 3′ single-strand DNA (ssDNA) by nucleolytic degradation of 5′ DNA ends, called 5′ to 3′ end resection [24,25,26]. The ssDNA is then bound by Rad51 recombinase that searches and copies from homologous templates across the DNA break [27]. NHEJ, instead, seals broken DNA by DNA ligase IV after the juxtaposition and alignment of chromosome ends [28]. Importantly, 5′ to 3′ end resection plays a pivotal role in the repair pathway choice by inhibiting NHEJ [29]. End resection also commits cells to HR and unconventional end joining called microhomology mediated end joining (MMEJ) [29]. The importance of DSB repair is further underscored by the findings that multiple human diseases leading to immune dysfunctions or cancer predisposition are the results of mutations in DSB repair genes [28].

*DNA damage response (DDR)*: DNA damage triggers complex and cell-wide responses encompassing cell cycle arrest, gene expression, chromatin remodeling, energy control, programmed cell death and autophagy, in addition to DNA repair, many of which are essential for the integrity of eye functions and the age-related disease progression [30]. Central to DDR lies two apical protein kinases, ataxia telangiectasia mutated (ATM) and ataxia telangiectasia and Rad3 related (ATR), which initiate a cascade of signal transduction through multiple kinases and the effector molecules and orchestrate DNA damage repair with cellular homeostasis [31]. Accordingly, the eye diseases often feature dysregulation of these multi-faceted DDR and impaired coordination of cellular physiology and functions in eye and optic nerve systems, albeit with the intact DNA repair.

*DNA damage and repair in mitochondria*: Emerging evidence suggests that mitochondrial DNA integrity and homeostasis is essential to the subset of age-related diseases including eye diseases [32]. Mitochondria is a cytoplasmic organelle, producing cellular energy by oxidative phosphorylation. Mitochondrial DNA is particularly vulnerable to DNA damage because oxidative phosphorylation could generate reactive oxygens and high energy electrons, capable of inflicting further DNA damage (reviewed in [33]). Evidence suggests that most DNA repair pathways also operate at mitochondria but with substantial differences exist to nuclear counterparts [34]. Nuclear DNA damage signaling is also intimately linked with mitochondrial integrity, function and aging [35]. The details of mitochondrial DNA damage repair and signaling are still emerging and are currently the subject of intense investigation. However, a few recent findings firmly established the causal effect of mitochondrial DNA integrity on the prevention and treatment of age associated diseases [32], warranting further analysis of this topic. To learn more about DNA damage repair and signaling in cancers and aging associated diseases, we would also like to highlight excellent reviews listed on each repair pathway for the readers for further reading.

## 3. Diabetic Retinopathy

Diabetic retinopathy (DR), the most prevalent microvascular diabetes complication, is a leading cause of blindness that impacts nearly 100 million people globally. Although DR is primarily characterized by vascular dysfunction and capillary nonperfusion, it is caused by both vascular dysfunction and neurodegeneration [36]. Indeed, neuropathy, indicated by the diminished electrical activity in electroretinograms and thinning inner retinal layer that includes ganglion and amacrine cells, is present even before apparent retinal ischemia. Patients with DR can also develop diverse conditions including macular ischemia, diabetic macular edema, preretinal hemorrhage, vitreous hemorrhage, and tractional retinal detachment, underscoring its complex pathologies impacting multi-organ failure. DR can be classified as nonproliferative or proliferative, based on extraretinal neovascularization and/or proliferation [37].

Growing evidence suggests that diabetes induces accelerated DNA damage by multiple means, which could explain some of its complex pathologies [38,39]. The results suggest that hyperglycemia is at least partially responsible for elevated DNA damage in diabetes (Figure 2). Indeed, hyperglycemia can trigger oxidative damage and single-strand breaks on cultured endothelium [40] and several cell types in vitro [41]. Transcriptomic analysis indicates that the level of BER gene expression including apurinic/apyrimidinic endodeoxyribonuclease 1 (APEX1) and *N*-methylpurine-DNA glycosylase (MPG) shows strong correlation with the protection from microvascular complications in DR patients [42]. High carbohydrate exposure also results in the depletion of NAD+ (nicotinamide adenine dinucleotide) compared to NADH (nicotinamide adenine dinucleotide hydrogen) and defective DNA double-strand break (DSB) repair. Accordingly, reconstituting DSB repair prevents fibrosis instigated by metabolic stress [43].

The elevated glucose levels in the blood are also linked to cellular senescence and DNA damage, which could be responsible for organ fibrosis in diabetes complications. Hyperglycemia can also induce DNA damage by advanced glycated end products (AGEs). In a mouse model of type 2 diabetes, there was an increase in the level of DNA advanced glycation end products (DNA-AGEs) in urine and tissue (liver and kidney) [44]. Alternatively, hyperglycemia induces high insulin levels to control blood glucose, with consequent hyperinsulinemia causing a significant increase in DNA damage in vitro, which coincided with the generation of reactive oxygen species (ROS). 8-oxo-7,8-dihydroguanine (8-oxoG) is a hallmark of oxidative DNA damage and a primary mutagenic intermediate of oxidative stress [45]. Among the diabetic patients, those with proliferative DR had significantly higher 8-oxoG levels than those with nonproliferative DR or without DR [46]. In support of this, antioxidants, IGF-1 receptors, insulin blockers, and a phosphatidylinositol 3-kinase inhibitor treatment reduces ROS [47]. Furthermore, free fatty acids (FFA), which could lead to insulin resistance and increases the risk of diabetes, can be responsible for mitochondrial DNA damage [48,49]. The results of human antioxidant studies remain controversial despite encouraging outcomes in vitro studies. However, some combined antioxidant therapies appear promising [50].

Surprisingly, DR tends to continue to progress despite strict control of glucose levels after prolonged hyperglycemia. Hyperglycaemic memory is also illustrated after sustained microvascular damage, indicated through the loss of retinal pericytes. It is suggested that the metabolic memory phenomenon and the mitochondrial DNA (mtDNA) damage by reactive oxygen species might be potentially responsible for prolonged progressive course of DR [51]. The mtDNA repair system does not function adequately in chronic, unlike acute, hyperglycemia. Peripheral blood mitochondrial DNA damage can serve as a biomarker for DR since rodents with DR had increased blood mtDNA damage with decreased copy numbers compared with diabetic rodents without retinopathy and nondiabetic individuals [52]. Prominent consequences of mtDNA damage are subnormal complex I and III with reduced membrane potential. This causes the positive feedback loop where hyperglycemia induces superoxide, which damages mtDNA, impeding the electron transport chain and resulting in superoxide overexpression [51]. Moreover, hyperglycemia increases mtDNA sequence variants in the displacement-loop (d-loop), which contains transcription and replication components [53].

Epigenetic modification also takes part in the pathogenesis of DR [54]. Hyperglycemia activates histone deacetylase (HDAC) and increases its expression in the retina and capillary cells. Hyperglycemia concurrently downregulates the activity of histone acetyl transferase (HAT) and inhibits the acetylation of histone H3. Diabetes-induced changes for HDAC and HAT expression persist even after the termination of hyperglycemia. This result suggests that the deacetylation of retinal histone H3 could contribute to the metabolic memory phenomenon and the pathogenesis of DR [55]. Long noncoding RNAs (LncRNAs) are noncoding transcripts longer than 200 nucleotides that can bind specifically to DNA, RNA, or proteins. Diabetes overexpresses several LncRNAs that can translocate in the mitochondria, such as LncMALAT1 and LncNEAT1, which are encoded in the nucleus and participate in mitochondrial homeostasis. High glucose aggravates LncMALAT1 and LncNEAT1 expression, impairing mtDNA and mitochondrial membrane potential [56].

In DR, mtDNA is hypermethylated with increased 5mC levels, particularly at the d-loop region. The inhibition of DNA methylation thus decreases diabetes-induced base mismatches in the d-loop [57]. The overexpression of the enzyme Mlh1, associated with polymerase γ, mitigated the sequence variants in endothelial cells and decreased respiration rates while increasing apoptosis [53]. During the diabetes control and complication trial (DCCT), there was a persistency of DNA methylation over time at key genomic loci associated with diabetic complications in type 1 diabetes patients [58]. Further analysis is required to define the role of DNA methylation, mismatch repair, and base mismatches in D-loop.

## 4. Age-Related Macular Degeneration

Age-related macular degeneration (ARMD) is the leading cause of vision loss in individuals over 55. ARMD is characterized by the progressive deterioration of photoreceptors and outer retinal layers and the buildup of macular deposits, drusen. ARMD pathogenesis involves lipid deposition, chronic inflammation, oxidative stress, and inhibited extracellular matrix maintenance [59]. ARMD can be classified as non-neovascular or neovascular based on the presence of choroidal neovascularization [60]. Although age is the primary risk factor, other factors contributing to the progression of ARMD include genetic susceptibility, diet, smoking, and cardiovascular status. The development of ARMD is accompanied by the loss of integrity in retinal pigment epithelium (RPE) cells, photoreceptors and choriocapillaris, which relies on peripheral RPE to replace damaged central RPE cells in the macula. However, with senescence, degenerated areas in the macula cannot be replaced or regenerated [61]. A senescent cell shares characteristics of cancer cells, such as more prominent DNA damage, especially DSBs and elevated DDR, and chromosome aberrations. Senescent cells also secrete inflammatory cytokines, matrix-remodeling proteases, growth factors and chemokines, for example through the CXCR2 protein, that contribute to low-grade inflammations and aging-associated changes [61,62].

Consistent with the role of DNA damage in ARMD pathologies, oxidative stress causes damage to the DNA of RPE cells and hinders DNA repair ability with age. Using neutral comet and pulsed-field gel electrophoresis assays, cells taken from ARMD patients display greater endogenous DNA damage but not the double-strand breaks. In ARMD patients, DNA base oxidative modification is greater than in the controls, as probed by DNA repair enzymes NTH1 (endonuclease III-like protein 1) and Fpg (DNA-formamidopyrimidine glycosylase). Furthermore, DNA repair is less effective in lymphocytes from ARMD patients, and these lymphocytes are highly sensitive to hydrogen peroxide and UV radiation [63]. Patients with exudative ARMD had elevated 8-oxoG levels compared to control individuals [61]. In most cases, the repair of 8-oxoG is initiated by the 8-oxoguanine DNA glycosylase (hOGG1) via the BER pathway [64]. If 8-oxoG escapes this process and replicative DNA polymerase misinserts adenine instead of cytosine opposite to 8-oxoG, an alternative pathway of BER can be activated with the MutY glycosylase homologue (MUTYH, hMYH) to remove adenine [65,66]. The genetic variability in the hOGG1 and hMYH genes may be associated with ARMD occurrence and progression in human studies [15]. For the prevention of ARMD progression, antioxidants including lutein, zeaxanthin and vitamin C and E are used contemporarily [67]. In addition, metformin, a common diabetic medication, can function as an antioxidant and anti-inflammatory agent. The use of metformin is associated with decreased risk of developing ARMD and is currently under investigation for clinical use [68].

In addition to BER defects, the sensitivity of RPE cells to blue and UV lights with subnormal DNA repair may promote the development of ARMD [63]. Consistent with the role of UV damage repair deficit in ARMD, the mutation in ERCC6 gene, the key factor in transcription-coupled NER (TC-NER) for UV lesion repair, cause Cockayne syndrome (CS), an autosomal recessive disorder characterized by severely impaired physical and intellectual development, clinically recognized by features including photosensitivity, pigmentary retinopathy, and retinal degeneration [69]. Moreover, single nucleotide polymorphism (SNP) of the G allele in ERCC6 C-6530>G is associated with a risk of ARMD development [70].

The dysfunction of DNA repair in mitochondria also contributes to the pathogenesis of ARMD [71]. More mtDNA lesions in RPE cells are from the macular region rather than the periphery, and mtDNA repair capacity is particularly impaired in the macular region as well. Puzzlingly, unlike aging, which only affects the common deletion region in the mitochondrial genome, mtDNA lesions significantly increase in all regions of the mitochondrial genome in ARMD patients, and this mtDNA damage is associated with ARMD progression [72]. Indeed, mtDNA damage and ARMD stage have a positively correlated relationship, while mtDNA repair capacity and ARMD stage have a negatively correlated relationship. In addition, more mitochondrial heteroplasmic mutations, which have two or more variants within the same cell, are present with ARMD [73]. One interpretation is that overactive initiation of DNA repair systems by alkylating agents can lead to retinal damage and blindness in mice through BER-initiating alkyladenine DNA glycosylase (AAG). Thus, balancing the degree of DNA damage and the repair ability may be essential to preserve retinal function [74].

## 5. Glaucoma

Glaucoma is a type of optic neuropathy characterized by the degeneration of ganglion cells, which is related to increased intraocular pressure (IOP). Thus, controlling this pressure, which is directed by the secretion and drainage of aqueous humor by the ciliary body through the trabecular meshwork and uveoscleral outflow pathways, through medication, laser, or surgery is the primary therapeutic strategy for glaucoma. However, open-angle glaucoma patients have prominent outflow resistance through the trabecular meshwork [75]. The trabecular meshwork is a complex, perforated three-dimensional structure in the extracellular matrix composed of trabecular meshwork cells (TMC) [76]. Various signaling pathways are involved in the pathogenesis of glaucoma, such as TGF-beta, MAP kinase, Rho kinase, BDNF, JNK, PI-3/Akt, PTEN, Bcl-2, Caspase, and Calcium-Caspain signaling. These signaling pathways converge into proapoptotic gene expression, suppression of neuroprotective and pro-survival factors, and fibrosis at the trabecular meshwork, which causes increased resistance to aqueous humor drainage and elevation of IOP [77]. IOP-induced mechanical stress can also result in the distortion and remodeling of the lamina cribrosa, leading to impaired axonal transport of essential trophic factors from relay neurons in the lateral geniculate nucleus to ganglion cells. In addition, during metabolic stress from high IOP, retinal ganglion cells may have difficulty producing sufficient energy because of mitochondrial dysfunction. The characteristic optic nerve head with a greater cup to disc ratio and decreased thickness of the retinal nerve fiber layer appearances of glaucoma develop with the death of retinal ganglion cells and optic nerve fiber. These changes, the most crucial aspect of a glaucoma diagnosis, are apparent in the visual field test [75].

One pathogenic factor of glaucoma development is oxidative DNA damage. The oxidatively modified DNA base 8-hydroxy-2′-deoxyguanosine (8-OHdG) is a marker of oxidative DNA damage [78]. 8-OHdG increases in aqueous humor and serum of glaucoma patients [79]. In open-angle glaucoma patients, the amount of 8-OHdG is significantly prominent in the trabecular meshwork and correlates positively with IOP and visual field deterioration [80]. With 8-week oral antioxidant supplementation, 8-OHdG can be reduced in glaucoma patients with relatively high oxidative stress [81]. Antioxidant supplementation in glaucoma patients may be a promising therapy [82]. Furthermore, BER is deficient in glaucoma patients [83]. The expression of poly (ADP-ribose) polymerase (PARP1) and 8-oxoguanine DNA glycosylase (hOGG1), the two key BER enzymes, is significantly decreased in glaucoma patient cells. PARP1 detects DNA damage and facilitates the repair process through uncondensed chromatic structures and interactions with multiple DNA repair factors. OGG1 removes the modified base by cleaving the glycosidic bond [78]. Furthermore, the 399 Arg/Gln genotype of the X-ray repair cross-complementing group 1 (XRCC1) gene is associated with poor DNA repair ability and is related to an increased risk of primary open-angle glaucoma (POAG) occurrence and progression [83]. POAG patients also exhibit a variety of mitochondrial abnormalities, and the accumulation of mtDNA damage is responsible for its pathogenesis. Increased mtDNA deletion is accompanied by reduced mitochondria count per cell and cell loss in POAG. mtDNA deletion is transferred to mitochondrial progeny, progressively increasing with age [84]. The mtDNA to nDNA ratio, representing the degree of mitochondrial DNA damage, is inversely correlated with impaired ocular blood flow in male patients with severe open-angle glaucoma [85]. In a whole-mitochondrial genome sequencing study, half of the POAG patients had pathogenic mitochondrial mutations, and 36.4% were in complex 1 mitochondrial gene [86].

Evidence suggests that glaucoma is associated with epigenetic change that alters histone acetylation and DNA methylation and modulate gene expression [77]. Acute optic nerve injury significantly increases histone deacetylase (HDAC) 2 and 3 transcripts and decreases histone H4 acetylation in retinal ganglion cells. Histone deacetylase inhibitors such as trichostatin A (TSA) and valproic acid reduce the loss of ganglion cells and even enhance axonal regeneration after optic nerve injuries. These suggest that abnormal histone acetylation/deacetylation may be related to retinal ganglion cell damage in glaucoma. Significant genomic DNA methylation has been found in peripheral monocular cells from patients and lamina crobrosa cells from human donors with open-angle glaucoma compared to a healthy control [87,88].

POAG is also associated with an increased number of DNA breaks in both the local trabecular meshwork and the systematic circulating leukocyte [89]. If double-strand breaks in neurons, the most harmful form of DNA damage, are not repaired appropriately, the persistent activation of the DNA damage response can cause dysregulation of the cell cycle and re-entry into G1 that leads to neural dysfunction, apoptosis, and senescence. In the early stage of double-strand breaks, the MRN (MRE11-RAD50-NBS1) complex, including the Mre11, Rad50 and Nbs1/Nbn proteins, activates the ataxia telangiectasia mutated (ATM) kinase for DNA damage responses such as cell-cycle arrest, repair, and apoptosis [90]. Another early response to double-strand breaks is the formation of γH2AX through the phosphorylation of the Ser-139 residue of the histone variant H2AX [91]. Compared to the control, laser-induced chronic glaucoma modeled in rhesus monkeys showed higher expressions of 8-hydroxyguanosine (8-OHG), which indicates oxidative stress, and γH2AX, which indicates DNA double-strand breaks, in the neurons of the lathheral geniculate nucleus (LGN), primary visual cortex (V1), and secondary visual cortex (V2). Apurinic/apyrimidinic endonuclease 1 (APE1) and DNA repair proteins Ku80, Mre11, Proliferating Cell Nuclear Antigen (PCNA), and DNA ligase IV were also elevated in the LGN, V1, and V2 [92]. The DNA damage response is important for neural development. Furthermore, persistent DNA damage response might be responsible for aging and neurodegenerative diseases such as Alzheimer’s disease and amyotrophic lateral sclerosis [93]. Clinically, optic nerve crush injury models are similar to glaucoma to the extent that retinal ganglion cell (RGC) death is a main pathologic phenomenon [94]. DNA damage responses can be attenuated through the inhibition of Mre11 in the MRN complex or ataxia telangiectasia mutated (ATM) kinase. Interestingly, attenuated DNA damage response after an optic nerve injury is also neuroprotective to retinal ganglion cells and promotes regeneration of their neurites [90]. Attenuating the DNA damage response pathway also promotes functional recovery after spinal cord injuries [90]. These results suggest that DNA damage response can contribute to glaucoma development, while promoting DNA stability and mutation prevention.

## 6. Conclusions

In vision-threatening conditions such as DR, ARMD, and glaucoma, DNA damage and repair is a relevant pathogenic mechanism that suggests a potential future direction for the prevention and therapy of these disabling conditions. In addition, it may hold value as an indicator of prognosis. Both mitochondrial and nuclear DNA damage and their associated repair mechanisms appear to be crucial. However, not all repair responses after DNA damage are beneficial, and a balance needs to be maintained. Hence, further research is required to better understand the role of DNA damage and repair in the progression of these disorders and to pave the way for development of new agents for better clinical outcomes.

## Figures and Tables

**Figure 1 ijms-24-03916-f001:**
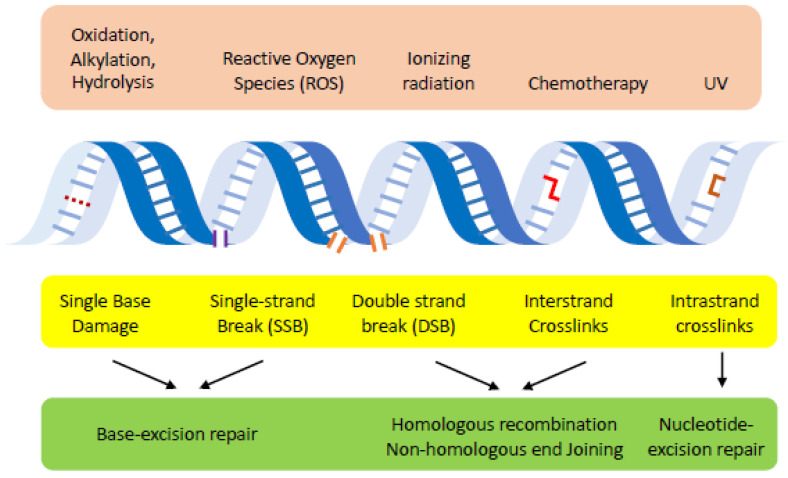
Overview of DNA damage and repair pathways. Environmental agents (i.e., radiation, chemotherapy and reactive oxygen species) or cellular metabolisms (oxidation, alkylation or hydrolysis) induce diverse types of lesions in the DNA. Specific pathways (base excision repair, nucleotide excision repair, non-homologous end joining and homologous recombination) are primarily responsible for repairing these lesions.

**Figure 2 ijms-24-03916-f002:**
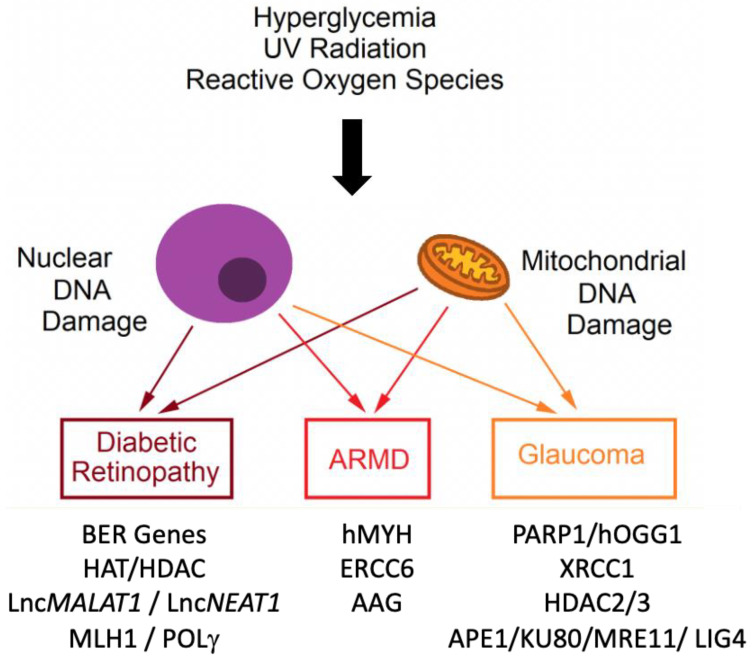
Hyperglycemia, ultraviolet radiation, and reactive oxygen species affect the development of glaucoma, DR, and ARMD (ARMD) via induction of mitochondrial and nuclear DNA damage. The list of DNA repair genes and pathways altered in each eye disease is shown as the molecular targets for therapy.

## Data Availability

Not applicable.

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
