# Peer review of "DNA Damage and Repair in Eye Diseases"

_ijms, 2023, doi:10.3390/ijms24043916_

Round 1
Reviewer 1 Report
In this review article, the authors have discussed the crucial role of damage and repair mechanisms of DNA in three different ocular pathologies including diabetic retinopathy, age-related macular degeneration, and glaucoma. Although the review provides enough details on the topic, the authors are requested to address the following comments.
1. Please provide abbreviations for diabetic retinopathy and age-related macular degeneration as DR and ARMD throughout the text except for the first mention.
2. On page 3, at the end of the base excision paragraph, the reference is missing.
3. Please avoid abbreviations like a.k.a. in the text.
4. Define ATM and ATR in the first mention. And check the abbreviations throughout the text to make sure they are defined in the first mention.
5. Please adhere to the instructions of the journal for adding references in the text. In some places, the references have been inserted as numbers, and in other places, numbers with PMIDs. Be consistent with adding references according to the style of IJMS.
6. In Figure 2, use a single arrow instead of multiple arrows to point out the pathologies as both nuclear and mitochondrial DNA damages involve in the development of DR, ARMD, and glaucoma.
7. In DR section, add information about proliferative and non-proliferative DR and the role of DNA damage in these diseases.
8. In ARMD section, include information about the difference between dry and wet ARMD, and the impact of DNA damage in these diseases.
9. Please use IOP as an abbreviation to denote intraocular pressure in the glaucoma section.
10. Provide examples of identified molecular targets that evolve DNA damage mechanisms in DR, ARMD, and glaucoma.
Author Response
The manuscript was revised according to the reviewer’s comments and provides a few genes and pathways implicated in the pathologies of eye diseases and DNA damage signaling/repair. I am very thankful to the reviewers and their helpful comments. By addressing their comments, I believe that the manuscript is now much improved and becomes suitable for publication at International Journal of Molecular Science. The revised part was marked by track changes and shown in blue.
Reviewer 1.
- Provide abbreviations for diabetic retinopathy and age-related macular degeneration as DR and ARMD throughout.
We revised the manuscript to use the abbreviations except for the first time mentioned.
- On page 3, the reference is missing.
We added the missing reference. We apologize this mistake.
- avoid abbreviations like a.k.a. in the text.
We removed "a.k.a." and other nonscientific abbreviations.
- Define ATM and ATR. Check abbreviations throughout to make sure they are defined in the first mention.
We defined ATM and ATR as requested and check the entire text to define any abbreviations when used first time.
- Be consistent with adding references according to the styles of IJMS.
We reformatted references to stay consistent throughout the text.
- In figure 2, use a single arrow instead of multiple arrows.
We changed the figure 2 to address the reviewer's suggestion.
- In DR section, add information about proliferative and non-proliferative DR and the role of DNA damage in these diseases.
We revised the manuscript to add the information about proliferative and non-proliferative DR and their association with oxidative stress on Page 4 and 5.
- In ARMD section, include information about the difference between dry and wet ARMD, and the impact of DNA damage in these diseases.
We added the information as the reviewer requested on page 6 and 7.
- Please use IOP as an abbreviation to denote intra ocular pressure.
We used the abbreviation IOP as the reviewer suggested.
- Provide examples of identified molecular targets that evolve DNA damage mechanisms in DR, ARMD, and glaucoma.
The manuscript was revised to provide these examples in each eye disease section as well as the list of genes and pathways identified as molecular targets of DNA damage in Figure 2.

Reviewer 2 Report
The review manuscript by Sohn et al, describes the mechanisms of DNA damage and repair in retinal diseases. The concept of non-inherited mutation has emerged as a potential strong driver of retinal diseases. The manuscript links DNA repair dysfunction with acquired mutations underlying the onset and development of retinal diseases.
1-The review could beneficiate of a better description regarding the role of some specific DNA repair defects in some specific retinal diseases.
2- All the different sections are not written with the same consistency. Some are very descriptive without details (the BER section for example).
3 - When available, some examples of the genes, molecular pathways and cell types affected in the eye after DNA repair defects would also be an interesting addition. It is present only in rare occasions.
4 – Figure 2 is not really necessary as it shows only the obvious. Maybe a schematic or table highlighting the relationship between DNA repairs defects -> gene(s) mutated -> biological processe(s) affected and the development of retinal disease(s) would have a bigger impact.
5 -Some typos are present in the text and references are missing (just annotated “ref” or with a PMID).
Author Response
The manuscript was revised according to the reviewer’s comments and provides a few genes and pathways implicated in the pathologies of eye diseases and DNA damage signaling/repair. I am very thankful to the reviewers and their helpful comments. By addressing their comments, I believe that the manuscript is now much improved and becomes suitable for publication at International Journal of Molecular Science. The revised part was marked by track changes and shown in blue.
Reviewer 2.
- The review could beneficiate of a better description regarding the role of some specific DNA repair defects in some specific retinal diseases.
To address the reviewer's suggestion, we revised the manuscript to further highlight the reported changes in DNA repair gene expression indicative of specific DNA repair defects in each eye diseases.
- Some are descriptive without details (BER section for example).
We revised the BER and other DNA repair sections to provide further details and their connections to eye diseases. However, we felt that it is necessary to stay concise in DNA repair sections to focus primarily on the sources of DNA damage and their contributions to the pathologies of eye diseases.
- Some of examples of the genes, molecular pathways and cell types affected in the eye after DNA repair defects.
Although accumulating evidence implicates DNA damage to age-related eye diseases, at present, it is difficult to link particular DNA repair defects to specific types of eye diseases beyond what we described in the reviews. The purpose of this review is to provide unbiased and broad review of this topic to stimulate the research and highlight the key questions in this area.
- Figure 2 is not really necessary as it shows only the obvious. Maybe a schematic or table highlighting the relationship between DNA repair defects, gene(s) mutated and biological processes affected and the development of retinal diseases.
We modified the figure 2 to list the genes and pathways implicated in the development and progression of each eye disease in response to the reviewer's comments.
- Some typos are present and references are missing.
We checked the entre document to correct any spelling errors and add missing references.
Reviewer 3 Report
The review entitled “DNA Damage and Repair in Eye Diseases” is a state of the art of the more significant results from studies regarding the known repair processes for DNA repair in the context of the most frequent eye diseases, cataracts, diabetic retinopathy, age-related macular degeneration, and glaucoma.
The review is very well organized and written in a scientific fashionable style.
I have just a few suggestions that could improve the manuscript.
- Abbreviations should be described when included in the text, e.i. ATM, ATR, nDNA, etc, also in the figures (e.i. Fig. 2, ARMD).
- Page 3, fourth paragraph, too long sentence, difficult to understand. I suggest authors to rewrite it “ …Importantly, 5' to 3' end resection, the key early step in HR, playing a pivotal role in the repair pathway choice by inhibiting NHEJ and committing cells to HR and unconventional end joining called microhomology mediated end joining (MMEJ)[25]”.
- From pg 4, references included in the text are linked to PMID, e.i. “[32, 33][PMID: 26021514, PMID: 20587728]”, is it a typo?
Author Response
The manuscript was revised according to the reviewer’s comments and provides a few genes and pathways implicated in the pathologies of eye diseases and DNA damage signaling/repair. I am very thankful to the reviewers and their helpful comments. By addressing their comments, I believe that the manuscript is now much improved and becomes suitable for publication at International Journal of Molecular Science. The revised part was marked by track changes and shown in blue.
Reviewer 3.
- Abbreviations should be described when included in the text.
We revised the entire documents to describe abbreviations when used for the first time.
- Page 3, fourth paragraph, too long sentence, difficult to understand. I suggest authors to rewrite it “…Importantly, 5’ to 3’ end resection, the key early step in HR, playing a pivotal role in the repair pathway choice by inhibiting NHEJ and committing cells to HR and unconventional end joining called microhomology mediated end joining (MMEJ) [25].
We rewrote the sentence to improve the readability.
- From pg4, references included in the text are linked to PMID, e.i. “[32, 33] [PMID: 26021514, PMID: 20587728]”, is it a typo?
We checked the entire document to correct any spelling errors and add missing references.
Round 2
Reviewer 2 Report
The review manuscript by Sohn et al, describes the mechanisms of DNA damage and repair in retinal diseases. I would like to thank the authors for improving the manuscript in particular regarding the genetic basis on retinal diseases associated with DNA repair dysfunction.
This review would be an interesting contribution to the field stating the current knowledge of this quickly evolving topic.
Minor issue: there are still some typo in the manuscript that should be corrected.